# Hawkeye: Model Collaboration for Efficient Reasoning

**Jianshu She[1][\*], Zhuohao Li[2][\*], Zhemin Huang[3], Qi Li[4], Peiran Xu[2],**
**Haonan Li[1], Qirong Ho[1]**
[1]Mohamed bin Zayed University of Artificial Intelligence (MBZUAI)
[2]University of California, Los Angeles
[3]Stanford University
[4]Independent Researcher

## Abstract

Chain-of-Thought (CoT) reasoning has demonstrated remarkable effectiveness in enhancing the reasoning abilities of large language models (LLMs). However, its efficiency remains a challenge due to the generation of excessive intermediate reasoning tokens, which introduce semantic redundancy and overly detailed reasoning steps. Moreover, computational expense and latency are significant concerns, as the cost scales with the number of output tokens, including those intermediate steps. In this work, we observe that most CoT tokens are unnecessary, and retaining only a small portion of them is sufficient for producing high-quality responses. Inspired by this, we propose HAWKEYE [‡], a novel post-training and inference framework where a large model produces concise CoT instructions to guide a smaller model in response generation. HAWKEYE quantifies redundancy in CoT reasoning and distills high-density information via reinforcement learning. By leveraging these concise CoTs, HAWKEYE is able to expand responses while reducing token usage and computational cost significantly. Our evaluation shows that HAWKEYE can achieve comparable response quality using only 35% of the full CoTs, while improving clarity, coherence, and conciseness by approximately 10%. Furthermore, HAWKEYE can accelerate end-to-end reasoning by up to 3.4× on complex math tasks while reducing inference cost by up to 60% [†].

## 1 Introduction

The emergence of reasoning-capable large language models (LLMs) (OpenAI et al., 2024; DeepMind, 2024; Guo et al., 2025) has recently made headlines. Equipped with Chain-of-Thought (CoT) reasoning(Wei et al., 2023), these models "think" by producing lengthy internal reasoning traces before generating a final response. This reasoning paradigm decomposes complex questions and applies multiple strategies to verify and refine answers, mimicking human-like problem solving. It has proven particularly effective for tasks that require fine-grained, step-by-step logical synthesis, such as mathematics and code generation (Zhang et al., 2022; Sprague et al., 2024).

However, such test-time compute scaling is inefficient, as a large portion of the generated "thinking" tokens is redundant. While long CoTs can improve reasoning quality, generating them during inference introduces significant overhead. For example, the OpenAI o1 model (OpenAI et al., 2024) uses up to 40K tokens, whereas GPT-4o averages around 4K tokens for the same query (Patel, 2024), leading to a 10× increase in KV cache memory usage. Moreover, post-training and deployment of reasoning models are extremely costly.

---

[\*] Equal contribution.
[‡] Project website: link
[†] HAWKEYE has been open-sourced at GitHub and models are available at Huggingface.
[§] Correspond to <jianshu.she@mbzuai.ac.ae>

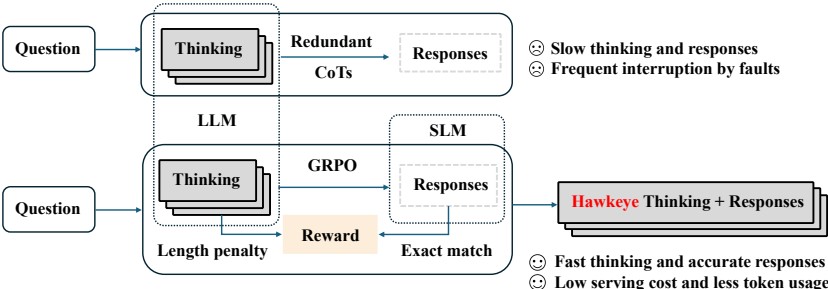

Figure 1: Overview of HAWKEYE and Comparison with Existing Reasoning Models. HAWK-EYE restructures the reasoning pipeline through model collaboration: a large model generates critical Chain-of-Thought (CoT) instructions, while a small model expands these instructions into complete responses to improve readability. The small model is fine-tuned using reinforcement learning on high-density CoTs. HAWKEYE enables fast, efficient, and cost-effective reasoning by reducing token usage without sacrificing response quality.

Knowledge distillation is often necessary to improve accessibility in reinforcement learning (RL)-based post-training. For example, the API cost of OpenAI o1 is approximately $6\times$ higher than that of GPT-4o. As reasoning models continue to scale at inference time, improving their efficiency has become an increasingly urgent challenge.

Although prior work has explored improving the test-time compute efficiency of reasoning, many approaches remain limited to self-reflection or multi-turn CoT tuning. For instance, Cheng & Durme (2024) reduce computational overhead by streamlining intermediate reasoning steps, while Wu et al. (2025) systematically examine the trade-off between efficiency and precision in large-scale reasoning. Additionally, Fu et al. (2024) introduce request tracking and scheduling based on model certainty, enabling early stopping to meet service-level objectives (SLOs) for latency-sensitive users. However, these methods do not address the fundamental challenge of overthinking in reasoning models: **How can we measure reasoning redundancy and effectively exploit it to enable more efficient thinking and generation?**

In this work, we propose HAWKEYE, a fast, efficient, and cost-effective CoT reasoning pipeline that is adaptable to a wide range of reasoning models. Our key insight spans both the inference and post-training stages. At inference time, we find that only the critical 20% of tokens meaningfully contribute to the final answer, while the remaining 80% introduce redundancy and can be safely removed. We provide a comprehensive assessment of this phenomenon across several popular reasoning models and diverse datasets. During post-training, we leverage reinforcement learning to guide models in identifying and retaining only the most informative tokens, enabling high-density reasoning. HAWKEYE introduces **model collaboration**, wherein a powerful model generates concise reasoning instructions, and small models expand them into human-readable responses. By retaining only critical CoTs and discarding redundant tokens during inference, HAWKEYE significantly reduces CoT reasoning complexity and cost, while maintaining—or even improving—response quality across datasets of varying difficulty levels (e.g., GSM8K (Cobbe et al., 2021), MATH (Hendrycks et al., 2021), MATH500 (Lightman et al., 2023), AIME). In our evaluation, HAWKEYE achieves up to a 67.6% reduction in reasoning tokens, a 62% reduction in serving cost, and a $3.4\times$ speedup in end-to-end reasoning compared to state-of-the-art reasoning models.

Our contributions can be summarized as follows:

1. We propose a novel reasoning paradigm, HAWKEYE, which utilizes model collaboration for efficient reasoning. In this setup, a large model generates concise reasoning instructions, which a small model then expands into full responses. HAWKEYE significantly reduces both computational and financial costs while preserving the re-

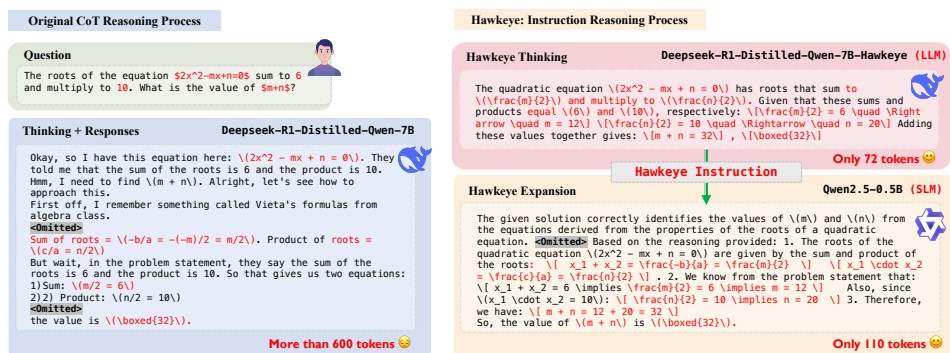

Figure 2: An example of HAWKEYE reasoning. The red highlights marks critical tokens that cannot be removed. HAWKEYE generates a high density of critical tokens and reduces token usage by over 70% (600 → 182 tokens). More examples can be found in Appendix C.

      sponse quality. We provide both empirical and theoretical evidence demonstrating the effectiveness of this collaborative reasoning framework.

2. We present the first systematic study of CoT redundancy, showing that excessive reasoning tokens are a widespread phenomenon across various tasks. Our analysis reveals that a substantial portion of these tokens can be removed using a principled compression strategy without degrading output quality.

3. We curate a high-quality CoT dataset and fine-tune a reasoning model using reinforcement learning to optimize CoT generation. This approach reduces CoT length by over 75% compared to the original large model while maintaining performance, with only a 4% drop in accuracy on the evaluated dataset.

## 2 Related Work

**Chain of Thought** Chain-of-Thought (CoT) reasoning has become a dominant approach in recent reasoning models such as OpenAI o1 (OpenAI et al., 2024) and DeepSeek-R1 (Guo et al., 2025). It generates rationales for questions and encourages models to think before responding. The quality of CoTs is critical, as errors can propagate due to the auto-regressive nature of generation. Several approaches have been proposed to optimize CoT reasoning, including self-consistency (Wang et al., 2023), least-to-most prompting (Zhou et al., 2022), Graph of Thoughts (Yao et al., 2023a), and ReAct (Yao et al., 2023b).

**Inefficiency in Reasoning Models** Despite the success of reasoning models, step-by-step thinking is often lengthy and incurs substantial computational overhead. To address this issue, recent studies have focused on improving reasoning efficiency. For example, Han et al. (2024) propose generating token budget-aware prompts for CoT reasoning. Building on this idea, Xu et al. (2025); Luo et al. (2025); Arora & Zanette (2025) introduce token limit hints to encourage more informative reasoning steps. Beyond token budget methods, Ma et al. (2025) identify a direction in the model parameter space that effectively controls the length of CoTs. Other research (Liu et al., 2024) explores reasoning shortcuts to further reduce unnecessary steps. However, these methods often rely on manually tuned hyperparameters and lack the capacity to automatically balance reasoning efficiency and answer accuracy.

**Reinforcement Learning in LLMs** Reinforcement learning (RL) has shown strong results in post-training reasoning models. Early work (Rafailov et al., 2023) aligned models with human feedback (RLHF) using classic algorithms like PPO (Schulman et al., 2017) and TRPO (Schulman et al., 2015). To improve scalability, methods such as DPO (Rafailov et al., 2023) and GRPO (Shao et al., 2024) reformulated alignment as a ranked contrastive loss over human preferences. Modern RL approaches avoid explicit value function estimation while preserving preference-aligned learning. DeepSeek-R1 (Guo et al., 2025)) demonstrated

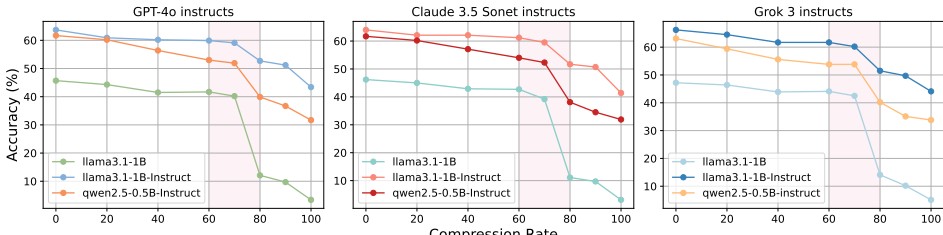

Figure 3: To examine redundancy in Chain-of-Thought (CoT) reasoning, smaller models (LLaMA3.1-1B, Qwen2.5-0.5B) were guided by larger models (GPT-4o, Claude 3.5 Sonnet, Grok 3) to generate CoTs for GSM8K. The CoTs were refined via LLM-assisted feedback by removing (1) repeated, (2) filler-like, and (3) overly fine-grained tokens. Compression rate is defined as the ratio of remaining to original token count. Results indicate that 60%–80% of CoT tokens are redundant across most models. See Appendix A for experimental details.

that large-scale RL post-training notably boosts reasoning performance over supervised fine-tuning (SFT).

## 3 HAWKEYE

In this section, we present the design details of HAWKEYE, our framework for efficient post-training and reasoning inference. The core idea behind HAWKEYE's post-training is to minimize redundancy in intermediate reasoning tokens and accelerate reasoning, thereby reducing cost while enhancing CoT quality. During inference, HAWKEYE introduces a novel paradigm in which a large language model (LLM) guides smaller language models (SLMs) to generate answers based on distilled CoTs. HAWKEYE achieves nearly a 50% reduction in cost and significantly faster inference without sacrificing response quality.

### 3.1 CoT Redundancy

Reasoning models such as DeepSeek-R1 and OpenAI o1 leverage intermediate Chain-of-Thoughts (CoTs) to decompose problems and produce more accurate responses on challenging tasks. However, the quality of these CoTs is critical, as errors can propagate due to the auto-regressive nature of generation. While techniques such as self-consistency (Wang et al., 2022), Tree-of-Thought (Yao et al., 2023a), Graph-of-Thought (Besta et al., 2024), and Skeleton-of-Thought (Ning et al., 2023) enhance CoT reasoning through various optimizations, they largely overlook CoT token efficiency — a factor that can significantly affect overall performance.

We observe that CoT reasoning often contains substantial redundancy due to (1) repeated hints, (2) filler phrases (e.g., "Well," "Let me double-check"), and (3) overly fine-grained steps. Prior work (Han et al., 2024; Ma et al., 2025; Liu et al., 2024) has reported similar findings. To further investigate this, we designed an experiment to visualize the redundancy, as shown in Figure 3. We selected several small models that lack strong reasoning capabilities and instructed them using CoTs generated by reasoning models. This setup ensures that the models themselves do not introduce bias into the experiment. We compressed the CoTs by selectively removing intermediate steps (note: the final answer is not provided in this experiment). Accuracy remains stable until the compression rate approaches 70% (highlighted in Figure 3). This indicates that nearly 70% of intermediate CoT tokens can be safely removed without significantly affecting accuracy. We further conducted experiments on various benchmarks and reasoning models, and observed consistent results (see Appendix A). By refining positive samples and removing redundant tokens, we are able to reduce token usage by nearly 70% while still producing high-quality responses. We refer to the retained tokens as logic-bearing tokens, as they typically convey logical or mathematical reasoning (Figure 2).

Our preliminary results demonstrate the significant redundancy present in CoT reasoning. However, leveraging this redundancy remains challenging due to two key issues: (1) identifying critical CoTs is difficult, as there is a lack of efficient metrics to quantify their importance; and (2) existing methods (Ma et al., 2025) are difficult to adapt for online serving, rendering real-time reasoning infeasible. To address these challenges, we fine-tuned a DeepSeek-R1-distilled Qwen-7B HAWKEYE model offline via reinforcement learning. We found that the model effectively streamlines its reasoning process by focusing on a minimal set of critical steps, and it exhibits strong generalization capabilities. A detailed discussion is provided in Section 3.2.

## 3.2 HAWKEYE: post-training

Guo et al. (2025) employs a rule-based GRPO approach for post-training, thereby endowing the R1 model with state-of-the-art reasoning capabilities. Inspired by this post-training paradigm, we employ GRPO to fine-tune a 7B model, enabling it to generate more concise and accurate CoT instructions.

As shown in Figure 3, we observe that CoT density achieves an optimal trade-off between compression rate and performance when the compression rate ranges from 0.6 to 0.8 relative to the original. This allows for a reduction in length while preserving the original information. Based on this observation, we set the starting point of the length penalty to 0.3 in our experiments and apply a quadratic penalty proportional to the number of tokens exceeding this threshold.

---

**Algorithm 1** GRPO fine-tuning of Model A for *compressed* Chain-of-Thought generation

---

**Require:** Policy model $\pi_\theta$ (Model A), frozen response model (Model B), training dataset $\mathcal{D} = \{(q_i, a_i, c_i^{\text{orig}})\}$, instruction prompt $s$, length penalty weight $\lambda$, target compression ratio $\alpha$ (set to 0.3)

1: **while** Model A not converged **do**
2:     **for** each mini-batch $\{(q_i, a_i, c_i^{\text{orig}})\}_{i=1}^{N} \subset \mathcal{D}$ **do**
3:         **for** each $(q, a, c^{\text{orig}})$ in mini-batch **do**
4:             $c \leftarrow \text{ModelA}(q)$
5:             $\hat{a} \leftarrow \text{ModelB}(q, c, s)$
6:             $P \leftarrow \lambda \cdot \max\big(0, \ \text{len}(c) - \alpha \cdot \text{len}(c^{\text{orig}})\big)^2$
7:             $R \leftarrow \text{EM}(\hat{a}, a) - P$
8:             $\theta \leftarrow \theta + \eta \nabla_\theta \log \pi_\theta(c \mid q) \cdot R$
9:         **end for**
10:     **end for**
11: **end while**
12: **return** Fine-tuned parameters $\theta$

---

DeepSeek-R1's reward model evaluates only the exact match between the generated output and the ground-truth answer, along with the length of the CoTs, without imposing explicit constraints on the generation process. Accordingly, HAWKEYE's fine-tuning framework incorporates a small model as the answer generator, with the exact match reward computed based on its final output.

We fine-tune Model A (policy $\pi_\theta$) using GRPO to generate CoT instructions. For each training pair $(q, a, c_{\text{orig}})$, Model A generates a CoT $c$, and a weaker, frozen Model B produces a response $\hat{a}$ based on $(q, c, s)$, where $s$ is a fixed instruction prompt. The reward is computed as

$$R = \text{EM}(\hat{a}, a) - \lambda \cdot \max\big(0, \text{len}(c) - 0.3 \cdot \text{len}(c_{\text{orig}})\big)^2,$$

where EM is the exact match score and $\text{len}(\cdot)$ denotes token count. The model is updated via policy gradient:

$$\theta \leftarrow \theta + \alpha \nabla_\theta \log \pi_\theta(c \mid q) \cdot R.$$

This design choice encourages the large model to generate CoTs that are not only truncated but also more coherent and interpretable, thereby enhancing the small model's ability to follow the reasoning and produce correct answers. As a result, this approach mitigates reward hacking, where the large model might otherwise bypass reasoning and directly output the answer. Empirical results, as shown in Figure 4, indicate that omitting the small model leads to an abrupt reduction in CoT length in pursuit of higher rewards, whereas incorporating the small model results in a more gradual and controlled decrease, aligning more closely with the intended reward structure.

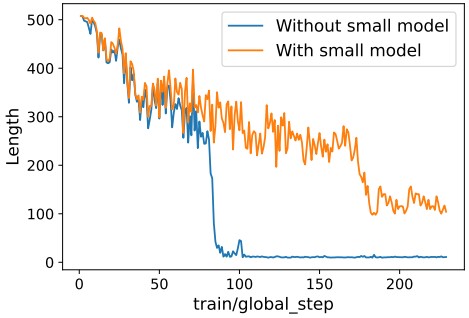 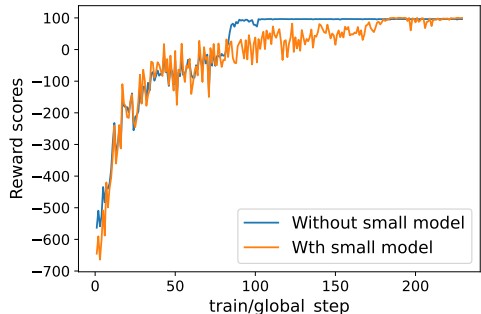

(a) CoT length across training steps. Without a small model, the policy collapses to extremely short CoTs in pursuit of higher rewards.

(b) Reward trajectory during training. Both converge to similar reward scores, but a small model can help avoiding reward hacking.

Figure 4: Effects of incorporating a small model during CoT generation. Including a small model stabilizes CoT length and mitigates reward hacking while achieving comparable reward scores.

### 3.3 HAWKEYE: inference

Drawing upon the findings presented in Section 3.1, we decompose the reasoning workflow of HAWKEYE into two phases. In the first phase, a large language model (LLM) constructs the core logical framework, thereby establishing a foundational structure for reasoning. In the second phase, a smaller language model (SLM) leverages this structured outline to generate comprehensive final responses. As illustrated in Figure 1, the SLM effectively elaborates on the distilled and logically rigorous reasoning traces (CoTs) derived from the LLM, ultimately producing complete and coherent answers.

Figure 2 provides a comparative analysis of two representative instances from HAWKEYE. The tokens explicitly highlighted in red represent essential components that encapsulate key reasoning steps, whose retention is critical to preserve reasoning fidelity. Empirical evaluations demonstrate that, through our reinforcement learning-enhanced instruction methodology, HAWKEYE achieves superior knowledge distillation compared to existing reasoning-based models.

Formally, our inference procedure can be articulated as follows:

$$x_{t+1} \sim \text{SLM}\left(x_{\leq t} \mid \text{prompt} = [q; c]\right)$$

In this formulation, $c$ denotes a validated chain-of-thought, recognized explicitly as a reliable reasoning trajectory. Consequently, the SLM is strategically guided to elaborate $c$ into a fully developed response.

Notably, the concept of two-step decoding using the same model was proposed as early as 2023 and has been demonstrated to be effective (Kojima et al., 2023; Lightman et al., 2023; Li et al., 2024). However, the question of how to obtain high-quality instructions for the second-step decoding remains underexplored. Existing studies predominantly rely on self-prompting strategies. In contrast, we explicitly introduce a dual-model decoding process during fine-tuning. As we demonstrate in the next section, our method improves both response quality (Figure 6) and throughput (Figure 7).

### 3.4 Experimental Setup

We build HAWKEYE based on PyTorch and perform evaluations on 8 NVIDIA H100 GPUs. Our baseline includes `DeepSeek-R1-Distilled-Qwen-7B`, a reasoning model capable of both reasoning and answering. We select MATH, MATH500, GSM8K, and AIME as our evaluation benchmarks. These benchmarks encompass complex reasoning tasks that span mathematics and commonsense reasoning.

## 4 Evaluation

### 4.1 Response Quality

HAWKEYE inference requires collaborative decoding between a large model and a small model. We evaluated the accuracy of `DeepSeek-R1-Distill-Qwen-7B` and HAWKEYE across various math datasets. As shown in Figure 5, HAWKEYE exhibits an accuracy drop ranging from 3% to 6% on MATH500, MATH, and GSM8K. However, once the large model generates the corresponding reasoning chain as an instruction——rather than directly providing a complete answer——the decoding process of the small model becomes more flexible, allowing the use of different system prompts or fine-tuning with specific datasets. This flexibility enables various enhancements, such as employing a safety-focused SLM to ensure responsible outputs or leveraging an RLHF-tuned model to better align with human preferences. In our experiments, we used `Qwen2.5-0.5B-Instruct` to generate the final response, optimizing for decoding efficiency.

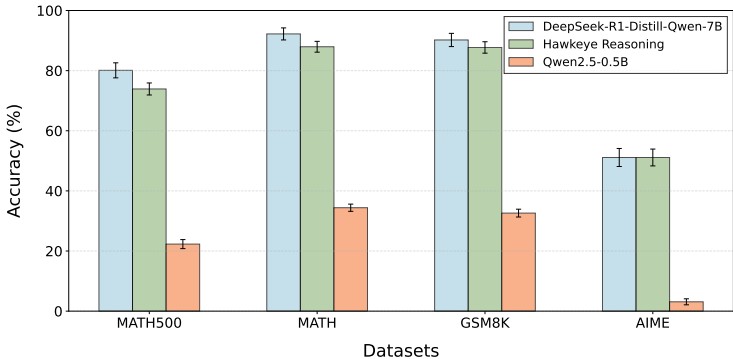

Figure 5: Exact match accuracy on MATH500, MATH, GSM8K, and AIME. HAWKEYE achieves comparable accuracy to the baseline while significantly reducing computational cost and token usage.

To assess whether the decoding process remains aligned with the original question and to assess the quality of responses generated by the 0.5B model, we conducted a comparative analysis using `GPT-4o`, `Grok-3`, and `Claude-3.5`. Specifically, we employed the LLM-as-Judge to compare the original outputs and those of the 0.5B model across five dimensions: *Coherence*, *Completeness*, *Clarity*, *Correctness*, and *Conciseness*. Results are presented in Figure 6, and detailed evaluation procedures are provided in Appendix B.

The results demonstrate that a more concise CoT strategy, combined with the 0.5B model, achieves performance comparable to that of the original model. Moreover, this approach exhibits superior performance in *Clarity*, *Conciseness*, and *Coherence*, highlighting its potential to enhance the user-friendliness of the decoding process.

### 4.2 System Latency

To rigorously assess the inference acceleration capabilities of HAWKEYE, we conduct an end-to-end latency evaluation, measuring the time span from initial user prompt processing to final token generation. Our benchmarking is implemented using SGLang (Zheng et al., 2024),

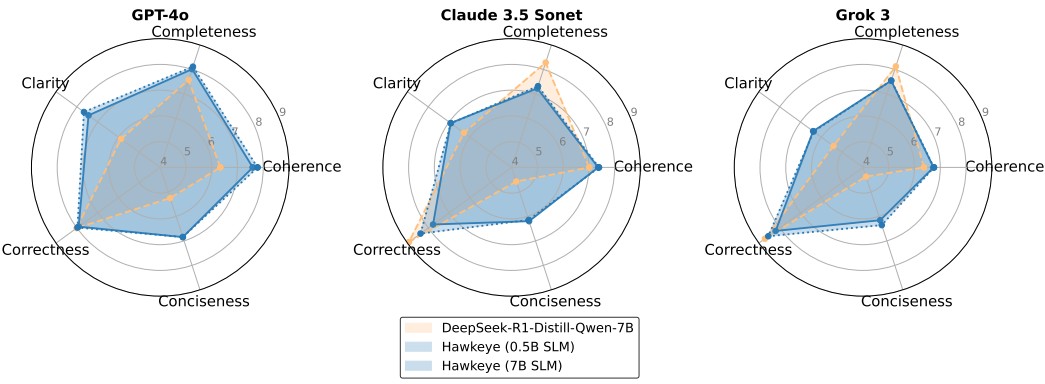

Figure 6: Response quality evaluated using LLM-as-Judge benchmarks (GPT-4o, Claude 3.5 Sonnet, Grok 3), scored across five aspects. Model collaboration is shown to improve response quality. Notably, increasing the size of the SLM (from 0.5B to 7B in our evaluation) offers limited additional benefit. See Appendix B for experimental settings.

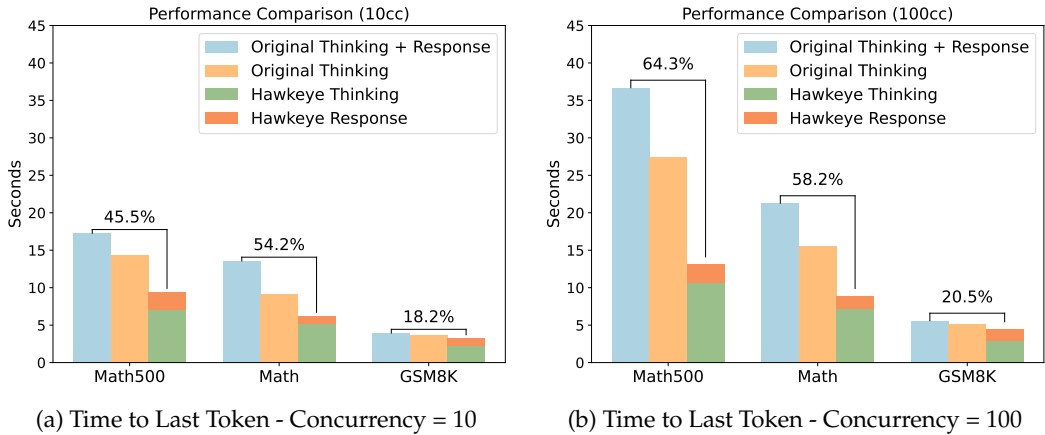

(a) Time to Last Token - Concurrency = 10    (b) Time to Last Token - Concurrency = 100

Figure 7: Time to last token under two concurrency levels (10 and 100) across different reasoning pipelines. **Original Thinking** refers to CoT generation by the DeepSeek-R1-Distill-Qwen-7B model, while **Hawkeye Thinking** utilizes compressed CoTs to improve token efficiency. Percentage labels indicate the latency reduction achieved by HAWKEYE. As concurrency increases, HAWKEYE's advantage becomes more pronounced (e.g., from 45.5% to 64.3% on MATH500), demonstrating better scalability by mitigating unnecessary reasoning.

which facilitates concurrent handling of multiple requests and enables precise recording of processing times for individual queries. Table 1 systematically presents the inference acceleration achieved by HAWKEYE across the complete reasoning pipeline. In the Hawkeye framework, the generated output can be decomposed into two components: the thinking tokens, which are enclosed within the special markers `[Think]` and `[/Think]`, and the response tokens. The "HAWKEYE (Full)" setting denotes the complete reasoning process, encompassing both the CoT reasoning sequence produced by the large model (i.e., the `[Think]` ... `[/Think]` segment) and the final response generated by the small model. In contrast, the "HAWKEYE (CoT-only)" setting considers only the large-model CoT reasoning tokens, excluding the subsequent response tokens.

Under a concurrency level of 10, HAWKEYE demonstrates inference speedups of up to 1.6×, 2.1×, and 2.5× relative to the baseline on GSM8K, MATH500, and MATH, respectively, while utilizing only 55.7%, 41.1%, and 38.3% of the original total token counts. When the concurrency is increased to 100, the speedup factors further improve to 1.8×, 3.4×, and 2.8×,

| Dataset | Method | Concurrency | Time per Req (s) ↓ | Avg TBT | Tokens ↓ |
|---|---|---|---|---|---|
| GSM8K | Baseline (Full) | 10 | 3.69 | 0.013 | 297.0 |
| | HAWKEYE (CoT only) | 10 | **2.37** | 0.014 | **165.7** |
| | HAWKEYE (Full) | 10 | **2.81** | 0.009 | 301.7 |
| | Baseline (Full) | 100 | 5.13 | 0.016 | 331.0 |
| | HAWKEYE (CoT only) | 100 | **2.93** | 0.017 | **171.0** |
| | HAWKEYE (Full) | 100 | **3.75** | 0.011 | 332.0 |
| Math500 | Baseline (Full) | 10 | 14.95 | 0.013 | 1136.54 |
| | HAWKEYE (CoT only) | 10 | **7.11** | 0.015 | **471.0** |
| | HAWKEYE (Full) | 10 | **8.31** | 0.011 | **771.4** |
| | Baseline (Full) | 100 | 36.20 | 0.021 | 1463.80 |
| | HAWKEYE (CoT only) | 100 | **10.74** | 0.022 | **474.9** |
| | HAWKEYE (Full) | 100 | **13.34** | 0.017 | **765.8** |
| Math | Baseline (Full) | 10 | 13.09 | 0.014 | 942.08 |
| | HAWKEYE (CoT only) | 10 | **5.14** | 0.014 | **361.23** |
| | HAWKEYE (Full) | 10 | **6.23** | 0.011 | **565.33** |
| | Baseline (Full) | 100 | 21.26 | 0.024 | 908.08 |
| | HAWKEYE (CoT only) | 100 | **7.48** | 0.022 | **332.50** |
| | HAWKEYE (Full) | 100 | **9.58** | 0.016 | **582.10** |
| AIME | Baseline (Full) | 10 | 93.48 | 0.016 | 5943.90 |
| | HAWKEYE (CoT only) | 10 | **66.60** | 0.016 | **4168.20** |
| | HAWKEYE (Full) | 10 | **68.90** | 0.015 | **4722.50** |

Table 1: Overall performance comparison across various datasets. *Time per Req* denotes average latency per request and *Tokens* denotes the average token count per request.

| Dataset | Method | Acc | Tokens | Compression Rate (%) |
|---|---|---|---|---|
| GSM8K | CoT-Valve (QwQ-32B-Preview) | 95.1% → 94.0% | 741.1 → 352.8 | 52.4 |
| | Hawkeye (DeepSeek-R1-Distill-Qwen-7B) | 90.7% → 88.9% | 331.0 → 171.0 | 48.3 |
| MATH | O1-Pruner (QwQ-32B-Preview) | 90.6% → 91.0% | 2191.0 → 1385.0 | 36.7 |
| | Hawkeye (DeepSeek-R1-Distill-Qwen-7B) | 92.3% → 89.4% | 942.0 → 361.0 | 61.7 |
| MATH500 | Hawkeye (DeepSeek-R1-Distill-Qwen-7B) | 80.1% → 75.5% | 1463.0 → 474.9 | 67.5 |

Table 2: Performance comparison between CoT-Valve, O1-Pruner, and Hawkeye across GSM8K, MATH, and MATH500 datasets. Compression Ratio is calculated as the percentage reduction in CoT tokens.

with corresponding token utilization reductions to 36.6%, 32.4%, and 36.6% of the baseline total tokens. See Figure 7 for detailed latency comparisons.

## 4.3 Cost Estimation and Saving

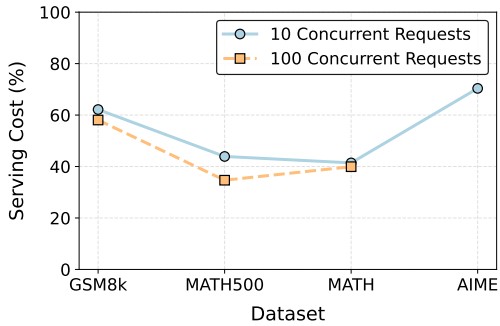

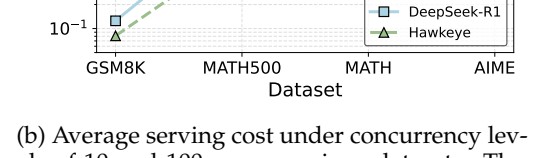

(a) Comparison of serving cost between HAWK-EYE and DeepSeek-R1, presented as a percentage scale.

(b) Average serving cost under concurrency levels of 10 and 100 across various datasets. The y-axis is shown on a logarithmic scale.

Figure 8: Serving cost for HAWKEYE

Let $C_{req}$ denote the cost per request, $T_{in}$ the number of input tokens, $P_{in}$ the price per input token, $T_{out}$ the number of output tokens, $P_{out}$ the price per output token, and $R$ the total number of requests. The serving cost is calculated as follows:

The cost per request is given by:

$$C_{req} = (T_{in} \times P_{in}) + (T_{out} \times P_{out})$$

The total cost $C_{total}$ is then:

$$C_{total} = C_{req} \times R$$

For the baseline comparison, we assume a fixed average input token count $T_{in}$ per dataset, with all requests being cache hits. Current API prices vary by model capability; for instance, OpenAI o1 has $P_{out} = \$60/\text{M}$ tokens, while DeepSeek-R1 has $P_{out} = \$2.19/\text{M}$ tokens. We similarly compute the serving cost of HAWKEYE and compare it with the baseline.

As the Figure 8 shows, due to the output token count saving, HAWKEYE costs less up to 98.40% and 59.09% than OpenAI-o1 and DeepSeek-R1. Notably, the AIME dataset only appears in the 10 concurrent requests since it doesn't have sufficient data for higher concurrency.

## 5   Conclusion

In this work, we introduce HAWKEYE, a novel paradigm for efficient reasoning via model collaboration. HAWKEYE enables a large model to generate concise and informative Chain-of-Thought (CoT) instructions, which are subsequently expanded by a smaller model to produce coherent and accurate responses. This collaborative reasoning framework significantly reduces redundancy in intermediate reasoning steps.

Extensive evaluations across multiple reasoning benchmarks (e.g., GSM8K, MATH500, AIME) demonstrate that HAWKEYE achieves comparable or even improved response quality, while reducing reasoning token usage by **50%–70%** and accelerating inference by up to **3×**. These results suggest that combining instruction distillation with small-model generation offers a promising pathway toward scalable and cost-effective reasoning systems without sacrificing performance.

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

# A    Appendix A

We use the following prompt to generate a CoT towards GSM8K dataset:

---

**Problem Prompt**

You are a helpful and thoughtful math tutor. Carefully work through the following problem step by step using clear reasoning and intermediate calculations. Explain your logic thoroughly, but **do not give the final answer**.
Problem:
¡Insert the problem here¿
Let's think step by step.

---

## A.1    Case Study

We first request the model to generate full Chain of Thought:

---

**Problem and CoT**

**Problem:** Rory orders 2 subs for $7.50 each, 2 bags of chips for $1.50 each, and 2 cookies for $1.00 each for delivery. There's a 20% delivery fee added at checkout, and she wants to add a $5.00 tip. What will her delivery order cost?
**CoT:**
To find the total cost of Rory's delivery order, we will break it down step by step. First, we calculate the cost of the subs. Rory orders 2 subs at $7.50 each, so the cost of the subs is:
$$\text{Cost of subs} = 2 \times 7.50 = 15.00$$
Next, we calculate the cost of the chips. Rory orders 2 bags of chips at $1.50 each, so the cost of the chips is:
$$\text{Cost of chips} = 2 \times 1.50 = 3.00$$
Then, we calculate the cost of the cookies. Rory orders 2 cookies at $1.00 each, so the cost of the cookies is:
$$\text{Cost of cookies} = 2 \times 1.00 = 2.00$$
Now, we calculate the subtotal. The subtotal is the sum of the costs of the subs, chips, and cookies:
$$\text{Subtotal} = 15.00 + 3.00 + 2.00 = 20.00$$
Next, we calculate the delivery fee. There is a 20% delivery fee added at checkout, so the delivery fee is:
$$\text{Delivery fee} = 0.20 \times 20.00 = 4.00$$
Finally, Rory wants to add a $5.00 tip. Therefore, the total cost including the tip is:
$$\text{Total cost with tip} = 20.00 + 4.00 + 5.00$$
At this point, the total cost of Rory's delivery order can be calculated.

---

Then we ask the model to preserve only some important tokens in CoT and compress the CoT to a certain level compared to the full CoT (80%, 60%, 40%, 30%, 20%, 10%).

---

**Prompt for Chain-of-Thought Revision**

You are given a detailed step-by-step reasoning (Chain-of-Thought) for solving a problem. Your task is to revise the explanation while keeping approximately (compression ratio)% of the original tokens unchanged. Make light edits for clarity or style if needed, but preserve the overall structure, meaning, and majority of the wording. Do not shorten or skip steps.

The key requirement is to **retain the most important information** in the original reasoning, ensuring that the core logic and steps are preserved.

Original CoT:

<Insert the original Chain-of-Thought here>

Revised CoT:

---

The compressed CoT should be:

---

**CoT Compressed to 80%**

Rory orders 2 subs at $7.50 each, totaling:
$$2 \times 7.50 = 15.00$$
2 chips at $1.50 each:
$$2 \times 1.50 = 3.00$$
2 cookies at $1.00 each:
$$2 \times 1.00 = 2.00$$
Subtotal:
$$15.00 + 3.00 + 2.00 = 20.00$$
Delivery fee (20% of subtotal):
$$0.20 \times 20.00 = 4.00$$
Add a $5.00 tip:
$$20.00 + 4.00 + 5.00$$
Total cost can now be computed.

---

**CoT Compressed to 60%**

2 subs: $15.00
2 chips: $3.00
2 cookies: $2.00
Subtotal:
$$15 + 3 + 2 = 20$$
Delivery fee:
$$0.2 \times 20 = 4$$
Tip: $5.00
Total is:
$$20 + 4 + 5$$

---

**CoT Compressed to 40%**

Calculate item costs: subs = $15, chips = $3, cookies = $2.
Subtotal = $20.
Add delivery fee (20%) = $4.
Add tip = $5.

---

**CoT Compressed to 30%**

Get subtotal from items: 15 + 3 + 2 = 20.
Calculate 20% delivery fee = 4.
Add tip = 5.

**CoT Compressed to 20%**

Item total = 20.
Fee = 4.
Tip = 5.

**CoT Compressed to 10%**

Items add up to 20.
Add delivery and tip.

## B  Appendix B

In evaluation of response quality, we use LLM-as-a-Judge to rate the response from clarity, completeness, coherence, correctness, and conciseness. We used the following prompt to rate the response:

---

**Prompt for Evaluating Math Response**

You are given a response to a specific math problem. Your task is to evaluate the response based on the following five criteria: clarity, completeness, coherence, correctness, and conciseness. Provide a score for each dimension based on the standards outlined below:

1. **Clarity (1–5):** - Does the response clearly explain each step of the reasoning? - Is the language used straightforward and easy to understand? - A score of 5 means the explanation is exceptionally clear with no ambiguity, while 1 means it is very difficult to follow.

2. **Completeness (1–5):** - Does the response address all parts of the question? - Does it include all necessary calculations and intermediate steps? - A score of 5 means the response covers all required steps and details, while 1 means critical parts of the problem are missing.

3. **Coherence (1–5):** - Are the steps and reasoning logically connected? - Does the response flow smoothly from one step to the next without any sudden jumps or gaps in logic? - A score of 5 means the response is well-structured and the reasoning is seamlessly connected, while 1 means the explanation is disjointed and hard to follow.

4. **Correctness (1–5):** - Does the reasoning lead to the correct result? - Are the intermediate steps logically sound and consistent with the problem statement? - A score of 5 means the solution is correct and matches the question exactly, while 1 means there are significant errors in reasoning or calculation.

5. **Conciseness (1–5):** - Does the response avoid unnecessary information or repetition? - Is it efficient without sacrificing important details? - A score of 5 means the response is brief yet thorough, while 1 means it includes excessive or irrelevant information.

Provide your evaluation below, giving a score from 1 to 5 for each criterion and explaining your reasoning behind the scores.

---

## C  Appendix C

As Figure 2 shows an example of the CoT comparison between Deepseek-distilled-qwen-7B model and HAWKEYE model, we show more examples in this sections.

---

**Question**

**Question:** Find the common ratio of the infinite geometric series:

$$\frac{-3}{5} - \frac{5}{3} - \frac{125}{27} - \cdots$$

---

**DeepSeek-R1-Distill-Qwen-7B Output (Full)**

To find the common ratio of an infinite geometric series, I know that each term is multiplied by a constant ratio, $r$, to get the next term.

So, let's denote the first term as $a_1$ and the second term as $a_2$. Then, the common ratio $r$ can be found by dividing the second term by the first term: $r = a_2/a_1$.

Given the series: $a_1 = -3/5$ and $a_2 = -5/3$.

Calculating $r$: $r = (-5/3)/(-3/5)$.

Dividing fractions involves multiplying by the reciprocal, so $r = (-5/3)*(5/-3) = 25/9$. But wait, 25/9 is greater than 1, which might be a problem because for an infinite geometric series to converge, the absolute value of the common ratio must be less than 1. Since $|25/9| > 1$, the series might not converge. However, the problem asks for the common ratio regardless of convergence, so $r = 25/9$. But let me double-check my calculations. First term: -3/5 Second term: -5/3 Third term: -125/27 Let me check the ratio between the second and third term:

$$r = (-125/27)/(-5/3) = (125/27)*(3/5) = (125 \times 3)/(27 \times 5) = 375/135 = 25/9$$

Yes, it's consistent. So the common ratio is 25/9. But wait, if the common ratio is 25/9, which is approximately 2.777..., that's a large ratio. Typically, in such problems, the ratio is a simple fraction like 2 or 3. Maybe I made a mistake somewhere. Let me recalculate: First term: -3/5 Second term: -5/3 .So, $r = a_2/a_1 = (-5/3)/(-3/5) = (-5/3)*(-5/3) = 25/9$. Wait, is that right? Let me verify: (-5/3) divided by (-3/5) is equal to (-5/3) multiplied by (5/-3). Wait, no, the reciprocal of (-3/5) is (-5/3). So, it's (-5/3) * (-5/3) = 25/9. Yes, that seems correct. So, the common ratio is indeed 25/9. But let me check the third term to see if it follows the same ratio. Third term should be $a_3 = a_2 * r = (-5/3)*(25/9) = (-125)/27$. Which matches the given third term. So, it is consistent. Therefore, despite the ratio being greater than 1, the common ratio is 25/9. But just to be thorough, let me consider if there might be a different interpretation. Alternatively, maybe the series is written with a typo or something, but as given, it's -3/5, -5/3, -125/27,... So, each term is being multiplied by (25/9) to get the next term. Therefore, the common ratio is 25/9. But I'm a bit surprised because 25/9 is a large ratio. Maybe I should express it as a mixed number, but 25/9 is already in simplest terms. Alternatively, maybe the problem expects the absolute value or something, but the question just asks for the common ratio. So, I think the answer is 25/9. But let me check once more: First term: -3/5 Multiply by 25/9: (-3/5)*(25/9) = (-75)/45 = (-5)/3, which is the second term.

Then, second term: (-5)/3 * 25/9 = (-125)/27, which is the third term. Yes, so it's consistent.

**Final Answer**

The common ratio of the infinite geometric series is $\boxed{\dfrac{25}{9}}$.

---

**HAWKEYE Output**

To find the common ratio of the infinite geometric series:

$$\frac{-3}{5} - \frac{5}{3} - \frac{125}{27} - \cdots$$

**Step 1:** Identify the first term ($a_1$) and the second term ($a_2$).

$$a_1 = \frac{-3}{5}, \quad a_2 = \frac{-5}{3}$$

**Step 2:** Use the formula for the common ratio ($r$):

$$r = \frac{a_2}{a_1} = \frac{\frac{-5}{3}}{\frac{-3}{5}} = \frac{-5}{3} \times \frac{5}{-3} = \frac{25}{9}$$

**Final Answer:**

$$\boxed{\frac{25}{9}}$$

---

**Question**

**Question:** The graph of the parabola defined by the equation

$$y = (x - 2)^2 + 3$$

is rotated 180 degrees about its vertex, then shifted 3 units to the left, then shifted 2 units down. The resulting parabola has zeros at $x = a$ and $x = b$. What is $a + b$?

**DeepSeek-R1-Distill-Qwen-7B Output (Full)**

**Question:** A parabola given by the equation $y = (x - 2)^2 + 3$ undergoes the following transformations:
- Rotated 180 degrees about its vertex, - Shifted 3 units left, - Shifted 2 units down. What is the sum $a + b$ of the zeros $x = a$ and $x = b$ of the resulting parabola? Alright, so I need to figure out this problem where a parabola is transformed in a few steps, and then I have to find the sum of its zeros. Let me try to break it down step by step.
First, the original parabola is given by the equation $y = (x - 2)^2 + 3$. I know that this is a standard parabola in vertex form, which is $y = a(x - h)^2 + k$, where $(h, k)$ is the vertex. So in this case, the vertex is at $(2, 3)$. Since the coefficient is positive, it opens upwards, which is the standard shape.
Now, the first transformation is rotating the parabola 180 degrees about its vertex. Hmm, rotating 180 degrees about the vertex... I think that would flip the parabola upside down. So, if it was opening upwards before, it will now open downwards after the rotation. How does that affect the equation?
I remember that rotating a graph 180 degrees around a point changes the direction of the parabola. So, in terms of the equation, that would mean changing the sign of the coefficient. The original equation is $y = (x - 2)^2 + 3$. If we rotate it 180 degrees about its vertex, the coefficient of the squared term will change from positive to negative. So, the new equation should be $y = -(x - 2)^2 + 3$.
Wait, let me make sure. If we rotate 180 degrees, it's like reflecting over the vertex point, which in this case is $(2, 3)$. So, yes, that would flip the parabola. So, the equation becomes $y = -(x - 2)^2 + 3$. Got that.
Next transformation is shifting the parabola 3 units to the left. Shifting left or right affects the x-coordinate of the vertex. Since it's 3 units to the left, we need to subtract 3 from the x-coordinate of the vertex. The original vertex was at $(2, 3)$. So, shifting left 3 units would bring it to $(2 - 3, 3) = (-1, 3)$.
How does this affect the equation? In the vertex form, shifting left by 3 units means replacing $x$ with $x + 3$. So, starting from $y = -(x - 2)^2 + 3$, replacing $x$ with $x + 3$ gives:

$$y = -((x + 3) - 2)^2 + 3 = -(x + 1)^2 + 3$$

So now the equation is $y = -(x + 1)^2 + 3$.
Next transformation is shifting 2 units down. Shifting down affects the y-coordinate of the vertex. The current vertex is at $(-1, 3)$. Shifting down 2 units would bring it to $(-1, 1)$.
In the equation, shifting down 2 units means subtracting 2 from the entire function:

$$y = -(x + 1)^2 + 3 - 2 = -(x + 1)^2 + 1$$

Now, I need to find the zeros of this resulting parabola. Set $y = 0$:

$$0 = -(x + 1)^2 + 1 \Rightarrow (x + 1)^2 = 1 \Rightarrow x + 1 = \pm 1 \Rightarrow x = 0 \text{ or } x = -2$$

Therefore, $a = 0$, $b = -2$, and $a + b = -2$.
**Final Answer:**

$$\boxed{-2}$$

> **HAWKEYE Output**
>
> To determine $a + b$ for the resulting parabola, follow these steps:
>
> 1. **Original Parabola:** - Equation: $y = (x - 2)^2 + 3$ - Vertex: $(2, 3)$
> 2. **Rotation:** - Rotating $180°$ about the vertex changes the equation to $y = -(x - 2)^2 + 3$.
> 3. **Shift 3 Units to the Left:** - Replace $x$ with $x + 3$: $y = -(x + 1)^2 + 3$.
> 4. **Shift 2 Units Down:** - Subtract 2 from the equation: $y = -(x + 1)^2 + 1$.
> 5. **Find Zeros:** - Set $y = 0$: $-(x + 1)^2 + 1 = 0 \Rightarrow (x + 1)^2 = 1 \Rightarrow x = 0$ or $x = -2$.
> 6. **Sum of Zeros:** - $a + b = 0 + (-2) = -2$.
>
> $$\boxed{-2}$$

