# OpenReview forum: "Hawkeye: Model Collaboration for Efficient Reasoning"
_colmweb.org/COLM/2025/Conference — COLM 2025_

### Official Review · Reviewer_9sDf · 2025-05-13

**Rating:** 6
**Confidence:** 3
**Ethics Flag:** 1

**Summary:**

The authors introduce HAWKEYE, a novel framework that improves the efficiency of Chain-of-Thought (CoT) reasoning in large language models (LLMs) by leveraging a collaborative paradigm between a large model (which generates concise reasoning instructions) and a small model (which expands these into full responses). The key insight is that most CoT tokens are redundant, and retaining only a fraction (about 20–40%) of the tokens can preserve answer quality while reducing computational cost. The framework uses reinforcement learning to fine-tune the large model to produce compact CoTs. Extensive evaluations show substantial reductions in latency (up to 3.4× speedup), token usage (50–70% fewer tokens), and serving cost, while maintaining or slightly improving response quality across multiple reasoning benchmarks.

**Reasons To Accept:**

1. The dual-model setup, where a large model generates compressed instructions and a small model completes the response, is novel and well-justified.

2. The use of GRPO with length penalties to produce dense, high-utility CoTs is methodologically sound and addresses reward hacking risks.

3. HAWKEYE is evaluated across four benchmarks (MATH500, MATH, GSM8K, AIME), demonstrating strong cost and latency improvements while maintaining accuracy.

**Reasons To Reject:**

1. While the small model reduces token costs, running two models' forward passes (one for instruction, one for response) might offset efficiency gains. This overhead is not explicitly measured or discussed.

2. The evaluation focuses heavily on math benchmarks. It would strengthen the paper to include more diverse reasoning tasks such as commonsense, multi-hop QA, or real-world decision-making scenarios.

---

> ### Author Response · Authors · 2025-06-03
> **Response to Reviewer 9sDf's Comments**
>
> We sincerely thank you for your review and constructive feedback! Below, we provide detailed responses to each of your comments.
>
> **1. Cost of two models' forward passes**: We sincerely thank the reviewer for raising this concern. Coordinating two models indeed introduces additional overhead, which mainly depends on the deployment scenario—particularly the communication cost and the small model's prefilling overhead.
>
> In **Table 1**, the reported latency under concurrent requests already includes the coordination cost of both models. Our experiments were conducted on the same machine within a single cluster, so the communication cost is negligible. The primary overhead comes from the small model’s prefill.
>
> When handling a single request, the cost of prefill and decoding for the small model is significantly lower than that of decoding with the large model:
>
> $
> T_{\text{SLM, prefill}} + T_{\text{SLM, decode}} \ll T_{\text{LLM, decode}}
> $
>
> For example, in our setting with SLM = 0.5B and LLM = 7B, the ratio
> `(T_SLM_prefill + T_SLM_decode) / T_LLM_decode`
> is roughly 0.2 to 0.3. As the size of the LLM increases, the efficiency gains become even more pronounced. We will add this explanation to the final version of the paper.
>
>
> **2. More benchmarks**: We indeed focused solely on mathematical reasoning in this work, as most state-of-the-art reasoning models currently concentrate on the math and code generation domains. These domains involve complex reasoning and provide standard answers, making them ideal for evaluating model reasoning capabilities.
>
> Our choice aligns with other works on efficient CoT modeling, which has similarly focused on math tasks. Nonetheless, we plan to extend our evaluation to include more  general domains in future versions to examine whether Hawkeye-trained models generalize beyond math.
>
> We hope our responses have addressed your concerns! :)

---

> > ### Comment · Reviewer_9sDf · 2025-06-10
> >
> > I want to thank the authors for their clarification. I'd like to keep my positive score.

---

> > > ### Author Response · Authors · 2025-06-11
> > >
> > > We sincerely thank you for your thoughtful review and for acknowledging that our responses addressed your concerns. We greatly appreciate your time and effort throughout the review process.

---

### Official Review · Reviewer_zrRu · 2025-05-13

**Rating:** 6
**Confidence:** 4
**Ethics Flag:** 1

**Summary:**

This paper proposes a CoT compression method based on GRPO and model collaborative optimization. During the model training phase, this approach encourages LLMs to generate concise and coherent CoTs, effectively mitigating the "reward hacking" phenomenon and improving reasoning reliability. In the inference phase, the LLM first constructs a core logical framework as the foundational structure for reasoning, which the SLM then utilizes to produce more comprehensive and refined final responses.

**Questions To Authors:**

1. Since the main datasets are centered on mathematical reasoning tasks, it is advisable to validate the proposed method's generalizability across a wider variety of tasks to confirm its effectiveness in more varied application scenarios.

2. Line 81 contains an incomplete sentence, which should be revised for clarity and completeness.

**Reasons To Accept:**

1. The authors conduct a quantitative analysis of the CoT redundancy problem, revealing that redundant reasoning tokens are a widespread and significant issue across various task scenarios.

2. The proposed method significantly reduces the length of CoT while effectively decreasing the model's inference latency, with only minimal performance degradation across multiple mathematical reasoning datasets.

**Reasons To Reject:**

1. It is commendable that the authors further investigate how different length penalty factors and dataset sizes affect the final experimental results to thoroughly evaluate the robustness and generalizability of their proposed method.

2. The current selection of baseline experiments seems somewhat limited. Conducting a more thorough comparison with existing reinforcement learning-based COT compression methods, such as [1][2], would help better demonstrate the relative advantages of the proposed approach.

[1] O1-Pruner: Length-Harmonizing Fine-Tuning for O1-Like Reasoning Pruning

[2] Training Language Models to Reason Efficiently

---

> ### Author Response · Authors · 2025-06-03
> **Response to Reviewer zrRu's Comments**
>
> We sincerely thank you for your review and constructive feedback. Below, we provide detailed responses to each of your comments.
>
> **1. Effect of length penalty factors and dataset size**: We evaluated different length penalty factors during training by varying the target compression rate and the results indicated a compression rate around 0.3 is a reasonable choice. Please refer to the visualization at: https://imgur.com/a/BoMxyca. When the compression rate is set below 0.3, GRPO tends to overly prioritize the reward associated with shorter CoTs, which in turn causes the model to neglect whether the small model can still produce the correct final answer. Nonetheless, we observed a consistent trend across different datasets and training scales, including both MATH and GSM8K. This suggests that the compression behavior is relatively robust to dataset size, at least in terms of guiding the model to produce more concise CoTs. We also talked about this in **Reward computation** point in kmZC's response.
>
> **2. Limited baselines**: (You can refer to Limited baslines" point we replied to rwTc's comments) We consider both CoT-Valve and O1-Pruner as concurrent works that have not undergone peer review. For this reason, we did not include them as baselines in the current paper. However, we will incorporate a detailed comparison with both methods in the next version. It is worth noting that both CoT-Valve and O1-Pruner are offline methods, whereas Hawkeye is an online approach that is lightweight and deployment-friendly.
>
> CoT-Valve requires a complex pipeline, including pairing long-form “standard CoT” chains with shorter variants, combined with LoRA or distillation to learn a compressed representation. This adds considerable complexity to the training process.
>
> O1-Pruner is based on PPO and is highly sensitive to hyperparameters such as the λ value, the number of samples  𝐾, and the PPO clipping range. It also relies on tens of thousands of training examples, making the process costly and less scalable.
>
> In contrast, Hawkeye uses GRPO-based online learning, requiring only 2K samples, a simple reward function, and smaller models. Despite the simplicity, it achieves better token savings and comparable or better compression ratios.
>
> For completeness, we provide a side-by-side comparison using the results reported in their drafts. While models and setups differ, we focus on compression rates and accuracy for fairness. We believe Hawkeye would outperform both methods under equal model scales.
>
> **CoT-Valve VS Hawkeye**:
> | Method | CoT-Valve | Hawkeye |
> |--------|-----------|---------|
> | Dataset | GSM8k | GSM8k |
> | Model | QwQ-32B-Preview | DeepSeek-R1-Distill-Qwen-7B |
> | Acc | 95.1% to 94% | 90.7% to 88.9% |
> | Tokens | 741.1 to 352.8 | 331 to 171 |
> | Compression Ratio (Higher is better) | 52.4% | 48.3% |
>
> **O1-Pruner VS Hawkeye**
> | Method | O1-Pruner | Hawkeye | Hawkeye |
> |--------|-----------|---------|---------|
> | Dataset | MATH | MATH | MATH500 |
> | Model | QwQ-32B-Preview | DeepSeek-R1-Distill-Qwen-7B | DeepSeek-R1-Distill-Qwen-7B |
> | Acc | 90.6% to 91.0% | 92.3% to 89.4% | 80.1% to 75.5% |
> | Tokens | 2191 to 1385 | 942 to 361 | 1463 to 474.9 |
> | Compression Ratio (Higher is better) | 36.7% | 61.7% | 67.5% |
>
> **Q1:** We indeed focused solely on mathematical reasoning in this work, as most state-of-the-art reasoning models currently concentrate on the math and code generation domains. We plan to include experiments in more general domains in future work to investigate whether a method trained on math also generalizes effectively to other tasks.
>
> **Q2:** We will make necessary formatting adjustments to improve it. We list the missed sentence of the 3rd point from the contribution part here for your reference: “We curate a high-quality CoT dataset and fine-tune a reasoning model using reinforcement learning to optimize CoT generation. This approach reduces CoT length by over 75% compared to the original large model while maintaining performance, with only around a 4% drop in accuracy on the evaluated dataset”.
>
> We hope our responses have addressed your concerns! :)

---

### Official Review · Reviewer_fxiX · 2025-05-16

**Rating:** 6
**Confidence:** 4
**Ethics Flag:** 1

**Summary:**

The paper proposes $\texttt{HAWKEYE}$, a 2-model framework designed to boost the throughput at inference time. The authors first highlight that existing CoT-based models are very redundant, and that the CoTs can be reduced by ~70\% in length without significantly impacting the performance. The authors propose a novel post-training technique, where the first model is trained to generate Concise CoTs, with length penalty is used to enforce conciseness. To enforce correctness, the resulting CoT is fed to a smaller model, which must then output the correct answer based on the initial model's concise CoT. The authors argue that this 2 model setup is less prone to collapse, where the model (due to the length penalty) ends up generating very short CoTs.

The authors evaluate on standard mathematical reasoning benchmarks, and also evaluate qualitatively (with LLM-as-a-judge) the generated CoT. Lastly, they present a detailed evaluation on model throughput, showcasing that their approach performs similarly as standard post trained models, but are much more efficient at inference time.

**Questions To Authors:**

1. It's unclear in Figure 3 what exactly the interplay between smaller and larger models. The CoT is generated by the smaller model, and then refined by the larger one ? or vice-versa ?
2. In In Figure 2, what is the purpose of the expansion step exactly, since it gives the same output as the initial instruction reasoning process
3. Regarding why Hawkeye's design mitigates reward hacking, "where the large model might otherwise bypass reasoning and directly output the answer", what is stopping the output of model A (c) to be just the answer (so c == solution). Wouldn't that achieve the optimal loss ?
4. In Figure 5, can you confirm that Deepseek-R1-Distill-Qwen-7B is used for Hawkeye's initial instruction as well ?
5. Minor typo : Line 81 is incomplete

**Reasons To Accept:**

1. The proposed method is original, flexible, and addresses a substantial bottleneck with reasoning based models
2. A novel analysis on the usefulness of long CoTs, and highlight the redundancy in standard CoT
3. The empirical analysis on model quality and throughput is well presented and well executed

**Reasons To Reject:**

1. While I agree that a reasonable amount of tokens can be removed from CoTs without impacting performance, I am sceptical about the 70% number. Looking at appendix A, we have
```
CoT Compressed to 20%
Item total = 20.
Fee = 4.
Tip = 5.
```
While the model could potentially get the right answer from this CoT, this does not mean that the model would have generated this CoT using the same token budget; in other words, to get to `item total = 20` for example, some computation had to be performed, and it's unclear whether the model would have gotten this right without first generating a longer version of the CoT.

2. It is still unclear to me whether the initial Hawkeye Thinking phase produces or not the answer. The text would suggest that it does not, however the example in Figure 2 suggests that it does. Therefore, it's unclear what role the 2nd LM serves. The authors mention that this 2 LM setup is flexible and could enable e.g. the SLM to be more safety aligned, but the paper does not showcase an example of this directly.

Overall, the paper would be stronger if the authors quantify how many times the first step generates the answer, and report performance for just the first step. Assuming it does generate the answer, the paper would be stronger if a) Figure 6 could be redone with and without the SLM, to show the qualitative impact of the SLM rephrasing.

---

> ### Author Response · Authors · 2025-06-02
> **Response to Reviewer fxiX's Comments -- Part 1**
>
> We sincerely appreciate your thoughtful review and constructive feedback! Below, we provide detailed responses to each of your comments.
>
> **1. Further explaination about token budget**: We agree with your observation. The example shown in Appendix A is indeed obtained by removing parts of the original CoT. The model would not necessarily generate this compressed CoT using the same token budget. Directly asking the model to produce a CoT with a token limit can reasily result in incorrect reasoning. However, our experiments results presented in Fig. 3 are not intended to imply that the model can naturally generate such a short CoT from scratch. We are intent to illustrate overthinking does exist and removing CoT redundancy in reasoning is empirically effective. In our reinforecement learning pipeline, shortening of CoTs is not performed in a single step, it is gradually—guided by the GRPO. The GRPO samples 64 candidate CoTs and selects the one with the highest advantage (i.e., a shorter CoT that still leads the small model to produce the correct answer) to update the model. Through this process, the model can gradually learn how to generate shorter CoTs from longer ones. It reduces the probability of producing incorrect CoTs when operating under limited token budgets, as it is not forced to immediately generate compressed CoTs without prior reasoning experience.
>
>
> **2. Initial Hawkeye thinking and 2nd LLM serves**: It is unpredictable whether the Hawkeye thinking could produce the correct answers or not. While Hawkeye thinking can potentially produce the final answers, it does not always do so explicitly. In some cases, the answer is embedded within the reasoning process. Our observations show that some Hawkeye thinking traces do not yield the correct answer directly, and responses without the SLM tend to be of lower quality. In such scenarios, the SLM serves an important role by retrospectively interpreting the thinking process to extract the correct answer and convert it into a more readable and user-friendly response. Furthermore, if the reasoning process terminates prematurely, is incomplete, or fails to arrive at a correct conclusion, the SLM can perform additional inference to recover or generate an appropriate answer (provide an extra turn to reasoning more). This flexible collaboration mechanism between two LLMs offers several practical advantages. For example, you can easily control the model's output by just controlling the SLM, not the large model. Fine-tuning the smaller model is easiser and flexible. If the final response needs to undergo safety screening, incorporate RLHF for improved readability, or follow specific stylistic or formatting constraints, these requirements can be fulfilled by fine-tuning only the 0.5B small model—without altering the reasoning capabilities of the full system.
>
> In summary, Hawkeye thinking plays the critical roles in heavy reasoning, whle Hawkeye SLM enables high flexibility and modularity, as the reasoning component remains consistent while the response generation can be easily customized. We are preparing an updated version of Fig. 6 that compares outputs with and without the SLM, highlighting its impact on improving response quality. It will be available soon and stay tuned!
>
> **Q1**:
>
> The workflow should be the opposite direction: the larger model is responsible for generating the CoTs, while the smaller model expands and rewrites the final response based on that.
>
> **Q2**:
>
> As we mentioned in our previous responses, the CoT should not be the final content delivered to users during inference, even though it contains answers sometimes, it still hard for users to quickly understand (not to mention sometimes it dosen't). Some existing LLMs even encode the CoT into formats that are not human-interpretable in order to make reasoning more efficient.
>
> Te purpose of the SLM can be summarized:
>
> * During training, it serves to prevent reward hacking by ensuring that the model optimizes for useful intermediate reasoning rather than simply maximizing a reward signal.
>
> * During inference, it converts the CoT into a more natural, human-friendly response.
>
> The expansion is thus not meant to generate a different answer, but to play a distinct role in shaping how the final output is presented. For additional discussion on the small model's utility, please refer to our detailed reply to Comment 2.

---

> > ### Comment · Reviewer_fxiX · 2025-06-05
> > **Response**
> >
> > Thank you for your response.
> >
> > 1. Regarding Figure 3 (coming back to points 1 and 2), how do you ensure that CoT generated by the larger model does not contain the answer ? If the goal is not to show that a model could generate the compressed CoT, but rather come up with the answer, it's crucial that the answer not be contained in the CoT.
> >
> > My other questions have been answered by the authors, thank you.

---

> > > ### Author Response · Authors · 2025-06-07
> > > **Response to Reviewer fxiX**
> > >
> > > We sincerely appreciate your active engagement in the discussion and your acknowledgment of our previous responses. Regarding the experiment presented in **Figure 3**, we would like to clarify the following:
> > >
> > > The CoTs used in Figure 3 were generated using high-capability models such as **GPT-4o**, **Claude**, and **Grok**. For these models, we employed a *self-prompting* strategy to iteratively compress their own CoTs into shorter versions with different compression ratios. As stated in Appendix A, Line 363, the prompts explicitly instructed the models **not to include the final answer** in their reasoning traces. Thanks to the strong instruction-following abilities of these models, we were able to reliably ensure that the resulting CoTs did not contain the answer.
> > >
> > > However, in actual deployment—such as in the scenario shown in Figure 2, where **Deepseek-R1-Distilled-Qwen-7B** is used to generate concise CoTs—the instruction-following capability is notably weaker compared to GPT-4 or Claude. This implies that even if we prompt the model not to include the answer, it may still generate CoTs that contain or imply the correct answer. This behavior reflects a characteristic of the Deepseek model itself.
> > >
> > > In conclusion, Figure 3 serves as a controlled experiment using strong models (GPT/Claude/Grok) to assess CoT redundancy. These models reliably follow instructions to exclude the final answer, allowing us to evaluate how much reasoning content can be compressed without compromising the SLM’s ability to derive the correct answer. The goal of having the SLM come up with the answer, given progressively shortened CoTs, is twofold: to demonstrate the presence of significant redundancy in full-length CoTs, and to validate the feasibility of a step-by-step token reduction process as a viable training strategy.

---

> ### Author Response · Authors · 2025-06-02
> **Response to Reviewer fxiX's Comments -- Part 2**
>
> **Q3**:
>
> In theory, the optimal loss would be achieved if the model could directly generate the correct answer as its CoT output (i.e., c=answer). However, in practice, this leads to unstable and brittle optimization, especially under GRPO training with a length penalty. In GRPO, the model samples multiple CoTs and selects the one with the highest advantage. Occasionally, a sampled CoT may skip the reasoning process entirely and still arrive at the correct answer. Due to its short length and correctness, this CoT receives a high reward and is used to update the model. However, this creates a dangerous optimization path: the model quickly learns to skip reasoning altogether to minimize token usage and maximize reward. Even though our GRPO is on a simple dataset. This could be a problem because for the majority of questions, reasoning is essential to arrive at the correct answer. Once the model starts omitting the reasoning step, it tends to reinforce that behavior across all subsequent generations. As a result, the model degenerates into producing short, ungrounded answers that frequently fail, which is the result of reward hacking.
>
> Our solution is to decouple reasoning and response generation, and compute the reward based on whether the small model (SLM), given the CoT, produces the correct answer. In this setup, if a CoT skips reasoning, the SLM is unlikely to recover the answer, and the model receives no reward. This discourages the model from exploiting lucky shortcuts and instead incentivizes it to include sufficient reasoning steps to support reliable downstream responses.
>
> **Q4**:
>
> Yes, you are correct.
>
> **Q5**:
>
> As we mentioned in a previous response, we thank for the reviewers to find out the typo and we will fix it in our next version. We list the missed sentence of the 3rd point from the contribution part here for your reference: *“We curate a high-quality CoT dataset and fine-tune a reasoning model using reinforcement learning to optimize CoT generation. This approach reduces CoT length by over 75% compared to the original large model while maintaining performance, with only around a 4% drop in accuracy on the evaluated dataset”*.
>
> We hope our responses have addressed your concerns! :)

---

### Official Review · Reviewer_kmZC · 2025-05-19

**Rating:** 7
**Confidence:** 4
**Ethics Flag:** 1

**Summary:**

This paper proposes Hawkeye, a novel approach where a large, expensive model is trained to generate succinct CoT chains that a smaller, cheaper model can then expand out to arrive at an answer. It shows that this approach (mostly) retains the solution quality while reducing the last LLM cost to about a half. Along with conciseness, the authors also find the clarity and coherence of CoTs is increased in this process. I found the approach interesting, the experiments well-thought-out, and the results quite convincing, and am generally inclined positively despite some weaknesses and points that need clarification.

**Questions To Authors:**

* In line 6 of the algorithm (also on text line 172), the reward is penalized with a term that is $\max(0, len(c) - 0.3 len(a))^2$. If I am interpreting correctly, this penalty term is non-zero only when $len(c)$, the CoT length, is at least 0.3 times $len(a)$, the response length. This choice is confusing, as I can't imagine a practical case where the CoT length is anywhere close to 0.3 times the response length. Isn't CoT length usually much larger than response length, i.e., $len(c) \gg len(a)$?

* Do you have a sense of why the plots in Fig 3 are so strikingly similar across 3 very different large LLMs producing the CoT? I can imagine these trends being similar across large LLMs, but these plots look really strongly close to each other, point by point, almost within minor fluctuations.

**Reasons To Accept:**

* Novel (to my knowledge) approach of using a small model to expand a succinct CoT generated by a large model.

* Systematic experiments motivating the approach by (a) showing that there is significant redundancy in CoT chains and (b) training a large LLM to generate concise CoTs usable by itself (as opposed to by a smaller model) doesn't work well due to reward hacking.

* Clear illustration of substantial reduction in inference cost while retaining performance (relative to the large model).

* Well-thought-out experiments to understand the proposed system.

**Reasons To Reject:**

* One could argue that cutting the cost in half is meaningful but not enough. E.g., half the cost of a large model is still very large compared to the cost of a small model. So, for deployed applications, training a smaller model if possible will remain more cost efficient.

* Some parts of the paper can use more clarity. E.g., to me, conciseness of the CoTs generated by the proposed method is the main selling point; coherence, clarity, etc., are secondary benefits. Yet, they are discussed on equal terms in section 4.1, especially around Fig 6. Similarly, in the experimental setup / method setup, it would be better to clearly state exactly what's the large LLM used and the small LLM used. From the first paragraph of sections 3.4 and 4.1, I *think* the large model is DeepSeek-R1-Distilled-Qwen-7B and the small model is Qwen2.5-0.5B-Instruct. If so, it would be good to state this clearly.

* Figure 7 leaves it unclear what happens if we use Hawkeye thinking + Original response. I think this would correspond to training the large model to generate a concise LLM (using the current setup), but then using the large model itself to process the concise LLM and derive an answer.

* Some design choices are unclear. E.g., see question below about reward computation.

---

> ### Author Response · Authors · 2025-06-01
> **Response to Reviewer kmZC's Comments -- Part 1**
>
> We sincerely thank you for your review and constructive feedback. We have carefully considered each of your comments and provide our detailed responses for each of them below:
>
> **1. Smaller model is better?**: It's ture that smaller models cost less, but their performance concerns still exist especially when directly train them via SFT or RL for reasoning. First, even if after distillation, the base model’s capacity ultimately determines the performance ceiling. For example, DeepSeek-R1-Distill-Qwen-1.5B is largely behind (~20% on AIME) DeepSeek-R1-Distill-Qwen-7B (https://huggingface.co/deepseek-ai/DeepSeek-R1). Second, in our evaluation, although Hawkeye uses a smaller model to generate the final response, it can still achieve comparable performance to a larger model (Fig. 5 and Fig. 6). We belive it provides evidence that our pipeline outperforms direct SFT or RL on smaller models. Additionally, directly fine-tuning small models introduces further challenges, such as a limited context window, which becomes a significant bottleneck when producing extended chains of thought. For relatively simple and common tasks (e.g. GSM8K, MMLU), small models are indeed sufficient to meet most practical needs. The limitations raise significantly when tackling more complex tasks.
>
> We also acknowledge that the reviewer’s observation is well-founded and aligns with this perspective.
>
> **2. clear declaration**: You are correct, the larger model we used is DeepSeek-R1-Distilled-Qwen-7B-Hawkeye, the smaller model is Qwen2.5-0.5B-Hawkeye. We will clarify this  explicitly in the next version, thank you for your suggestions!
>
> **3. Further explaination on Hawkeye thinking**: Hawkeye thinking refers to the concise CoT generation trained with our pipeline (the [Think]...[/Think] content produced by the GRPO post-trained model). In contrast, original thinking corresponds to the long-form CoT generation from the original model without Hawkeye. Since the model has been explicitly trained to treat the [Think]...[/Think] segment as the reasoning component, combining Hawkeye thinking with the original model's response would likely incur additional computational cost while offering limited benefit. This is because the original model is optimized primarily for strong reasoning capabilities, but not necessarily for generating concise or high-quality final responses. A key feature of our method is response generation, where we explicitly disaggregate reasoning ([Think]...[/Think]) from the final response and delegate the response generation task to a smaller model. This design not only reduces inference cost but also preserves quality in the user-facing output.
>
> In fact, there are specific reasons why we do not use the original model’s response in this setting. This choice reveals a potentially important insight: models that undergo extensive reasoning-focused post-training—especially on tasks like code generation and mathematical problem solving, which dominate current reasoning benchmarks—may exhibit degraded performance on other natural language tasks such as summarization or text rewriting. As a result, we argue that a smaller model, not overly optimized for reasoning tasks, may be better suited for generating the final user-facing response (since the CoT does not necessarily need to be presented to the user). Our results in Figure 6, which compare different models on response generation, could support this claim. However, we emphasize that this claim currently reflects our hypothesis. The extent to which reasoning-focused post-training causes catastrophic forgetting in other natural language tasks remains an open question and requires further empirical investigation.
>
> **4. Reward computation**: The value of 0.3 is selected based on the analysis in Fig. 3, where we observed that most models exhibit approximately 70% redundancy in their CoT. This value is treated as a hyperparameter. The 0.3 ratio refers to the **proportion of generated thinking tokens relative to the original CoT tokens**, not relative to the final response length.
>
> The reward should be computed as:
>
> $
> R = \text{EM}(\hat{a}, a) - \lambda \cdot \max\left(0, \text{len}(c) - \alpha \cdot \text{len}(c_{\text{orig}})\right)^2
> $
>
> Where:
> - $\hat{a}$: the predicted answer
> - $a$: the ground-truth answer
> - $c$: the generated (compressed) CoT
> - $c_{\text{orig}}$: the original, uncompressed CoT
> - $\alpha$ (set to 0.3): the target compression ratio
> - $\lambda$: the penalty weight
>
> We will revise the text and notation in the paper to clearly convey this definition and avoid further ambiguity.
>
> **We appreciate your time in reading our rebuttal and look forward to further discussions.**

---

> > ### Comment · Reviewer_kmZC · 2025-06-09
> > **Re: Response to Reviewer kmZC's Comments -- Parts 1 & 2**
> >
> > Thank you for the detailed explanation, which helped clarify most of my concerns.  Please include your explanations (concisely) in the revised paper, along with general writeup improvements as also asked by other reviewers.
> >
> > Assuming the authors take the time to make these suggested changes, I am happy to see this paper accepted!

---

> > > ### Author Response · Authors · 2025-06-10
> > > **Thank you!**
> > >
> > > Thank you sincerely for your kind words and constructive feedback. We are truly grateful for your thoughtful review and the time you took to engage with our work. Your recognition means a great deal to us, and we will carefully incorporate your suggestions and clarifications in the revision. Your comments have greatly helped us improve both the clarity and quality of the paper!

---

> ### Author Response · Authors · 2025-06-01
> **Response to Reviewer kmZC's Comments -- Part 2**
>
> **1. Reward argument selection**: please refer to **4. Reward computation** in our last response
>
> **2. Fig.3 looks similar**: This is a very good question, the curves in Fig. 3 are indeed similar. We observe these three LLMs share similar core reasoning logic. They differ only slightly in stylistic expression. This similarity may stem from the relatively simple and straightforward nature of GSM8K questions, which limits the observable variation between full and compressed CoTs. We anticipate that more complex reasoning tasks may reveal clearer distinctions but the trends could be similar in compression behavior. We hope these could address your concerns :)
>
> We provide one of our CoT breakdowns for Fig. 3 (also avaliable in Appendix A.1):
>
> *Qwestion*:
>
>     Problem: Rory orders 2 subs for $7.50 each, 2 bags of chips for $1.50 each, and 2
>     cookies for $1.00 each for delivery. There’s a 20% delivery fee added at checkout,
>     and she wants to add a $5.00 tip. What will her delivery order cost?
>
> *Claude*
>
>     I'll work through this delivery order problem step by step, calculating each component carefully.
>
>     Step 1: Calculate the cost of the subs
>     Rory orders 2 subs at $7.50 each, so I need to multiply:
>     Cost of subs = 2 × $7.50 = $15.00
>
>     Step 2: Calculate the cost of the chips
>     She orders 2 bags of chips at $1.50 each:
>     Cost of chips = 2 × $1.50 = $3.00
>
>     Step 3: Calculate the cost of the cookies
>     She orders 2 cookies at $1.00 each:
>     Cost of cookies = 2 × $1.00 = $2.00
>
>     Step 4: Find the subtotal of food items
>     Now I'll add up all the food costs to get the subtotal before fees:
>     Subtotal = $15.00 + $3.00 + $2.00 = $20.00
>
>     Step 5: Calculate the delivery fee
>     There's a 20% delivery fee added to the subtotal. I need to find 20% of $20.00:
>     Delivery fee = 20% × $20.00 = 0.20 × $20.00 = $4.00
>
>     Step 6: Add the tip
>     Rory wants to add a $5.00 tip on top of everything else.
>
>     Step 7: Determine what gets added together for the final total
>     The final cost will include:
>     - The food subtotal: $20.00
>     - The delivery fee: $4.00
>     - The tip: $5.00
>
>     So the calculation for the total would be: $20.00 + $4.00 + $5.00
>
> *GPT-4o:*
>
>     To find the total cost of Rory’s delivery order, we will break it down step by step.
>     First, we calculate the cost of the subs. Rory orders 2 subs at $7.50 each, so the cost of the subs is:
>     Cost of subs = 2 × 7.50 = 15.00
>
>     Next, we calculate the cost of the chips. Rory orders 2 bags of chips at $1.50 each, so the cost of the chips is:
>     Cost of chips = 2 × 1.50 = 3.00
>
>     Then, we calculate the cost of the cookies. Rory orders 2 cookies at $1.00 each, so the cost of the cookies is:
>     Cost of cookies = 2 × 1.00 = 2.00
>
>     Now, we calculate the subtotal. The subtotal is the sum of the costs of the subs, chips, and cookies:
>     Subtotal = 15.00 + 3.00 + 2.00 = 20.00
>
>     Next, we calculate the delivery fee. There is a 20% delivery fee added at checkout, so the delivery fee is:
>     Delivery fee = 0.20 × 20.00 = 4.00
>
>     Finally, Rory wants to add a $5.00 tip. Therefore, the total cost including the tip is:
>     Total cost with tip = 20.00 + 4.00 + 5.00
>
>     At this point, the total cost of Rory’s delivery order can be calculated.
>
> *Grok3:*
>
>     Step 1:  Rory orders 2 subs, and each sub costs 7.50. To find the total cost for the subs, multiply the price per sub by the number of subs:
>     2 × 7.50 = 15.00
>     So, the subs cost $15.00.
>
>     Step 2:  Rory orders 2 bags of chips, with each bag costing 1.50. Multiply the price per bag by the number of bags:
>     2 × 1.50 = 3.00
>     The chips cost $3.00.
>
>     Step 3: Rory orders 2 cookies, and each cookie costs 1.00. Multiply the price per cookie by the number of cookies:
>     2 × 1.00 = 2.00
>     The cookies cost $2.00.
>
>     Step 4: The subtotal is the sum of the costs for the subs, chips, and cookies:
>     15.00 + 3.00 + 2.00 = 20.00
>     The subtotal for the items is $20.00.
>
>     Step 5: The problem states there’s a 20% delivery fee added at checkout. A 20% delivery fee means 20% of the subtotal. To compute this, convert 20% to a decimal (20% = 0.20) and multiply by the subtotal:
>     0.20 × 20.00 = 4.00
>     The delivery fee is $4.00.
>
>     Step 6: Add the delivery fee to the original subtotal to get the total before the tip:
>     20.00 + 4.00 = 24.00
>     The total before the tip is $24.00.
>
>     Step 7: Rory wants to add a 5.00 tip:
>     24.00 + 5.00

---

### Official Review · Reviewer_rwTc · 2025-05-27

**Rating:** 6
**Confidence:** 4
**Ethics Flag:** 1

**Summary:**

This paper proposes HAWKEYE, a post-training and inference framework designed to improve the efficiency of Chain-of-Thought (CoT) reasoning in large language models (LLMs). HAWKEYE introduces a collaborative model architecture, where a large model generates concise CoT instructions, which are then expanded into full answers by a smaller model. The goal is to reduce token usage and inference latency while maintaining answer quality. To achieve this, the authors first analyze redundancy in CoTs and show that a large portion of tokens can be removed without significantly affecting performance. They use reinforcement learning (specifically GRPO) to fine-tune a generator model that outputs compressed but informative CoTs. During inference, these instructions guide a smaller model to produce coherent and accurate responses. The method is evaluated on multiple benchmarks (GSM8K, MATH, MATH500, AIME) and compared to existing models such as DeepSeek-R1.

**Questions To Authors:**

Please refer to the section "Reason to reject."

**Reasons To Accept:**

- The paper provides comprehensive experimental evaluations of the proposed framework, covering efficiency, accuracy, and the quality of CoT reasoning.
- The core idea of the paper is easy to follow.

**Reasons To Reject:**

- The writing quality of the paper requires further improvement. There are many pieces shown in the paper that lack any explanation. For instance, the contribution summary (Line 81) contains incomplete paragraphs. Additionally, the caption of Figure 2 does not align with the figure content, there is no example with 182 tokens as described.
- The proposed framework appears to cause a considerable performance drop (more than 2%) after compressing the CoTs, which raises concerns about its practical effectiveness. It would be helpful if the paper could showcase the actual accuracy numbers of the proposed framework to further clarify.
- The paper lacks comparison with other efficient reasoning baselines, such as CoT-Valve [1] (a supervised fine-tuning-based method) and O1-pruner [2] (a reinforcement learning-based method). Including these baselines would strengthen the evaluation and contextualize the effectiveness of the proposed approach.
- The distinction between HAWKEYE (Full) and HAWKEYE (CoT-only) is unclear. The paper presents both results in Table 1 but does not explain the differences between the two settings. If they represent different configurations, I am particularly interested in understanding the respective accuracy drops of HAWKEYE (Full) and HAWKEYE (CoT-only).

[1] Ma, Xinyin, et al. "CoT-Valve: Length-Compressible Chain-of-Thought Tuning." arXiv preprint arXiv:2502.09601 (2025).
[2] Luo, Haotian, et al. "O1-Pruner: Length-Harmonizing Fine-Tuning for O1-Like Reasoning Pruning." arXiv preprint arXiv:2501.12570 (2025).

---

> ### Author Response · Authors · 2025-06-01
> **Response to Reviewer rwTc's Comments**
>
> We sincerely thank the reviewer for the constructive feedback. Below, we address each of the concerns you raised.
>
> **1. Missed part on the section of contribution & Fig.2 caption**: Thank you for your feedback regarding the paper's formatting and typo. We will make necessary formatting adjustments to improve it. We list the missed sentence of the 3rd point from the contribution part here for your reference: *“We curate a high-quality CoT dataset and fine-tune a reasoning model using reinforcement learning to optimize CoT generation. This approach reduces CoT length by over 75% compared to the original large model while maintaining performance, with only around a 4% drop in accuracy on the evaluated dataset”*. The captions of Fig.2 is indeed align. The 182 tokens is calculated by summing thinking tokens (72 tokens) and response tokens (110 tokens), they are both involved in Hawkeye (full).
>
> **2. Showcase the actual accuracy numbers**: Fig. 5 can actually demonstrate there is no or minor accuracy loss in practice. We also verify the similar situation in our real-world deployment. We further provide more evaluation results on CNMO and HMMT, two recent math datasets (https://huggingface.co/deepseek-ai/DeepSeek-R1-0528). The accuracy numbers are shown in the table below:
>
> | Datasets| DeepSeek-R1-Distill-Qwen-7B | Hawkeye Reasoning | Qwen2.5-0.5B |
> | -------------- | --------------------------- | ----------------- | ------------ |
> | MATH500| 80.1% ± 3%| 75.5% ± 2%| 22.3% ± 1%|
> | MATH| 92.3% ± 2%| 89.4% ± 2%| 34.2% ± 1%|
> | GSM8K| 90.7% ± 2%| 88.9% ± 2%| 32.6% ± 1%|
> | AIME 2025 | 51.1% ± 2%| 51.1% ± 2%| 4.4% ± 2%|
> | CNMO 2025| 44.1% ± 1% | 40.7% ± 1%| 2%|
> | HMMT 2025| 40% ± 1%| 33.3%| 0%|
>
>  Due to the small size of the HMMT (only 30 questions), a single incorrect answer can impact the overall accuracy significantly, thus the accuracy on it exhibits a relatively large drop (7%). This can be eliminated by multi-turn reflexion.
>
> **3. Limited baselines**: We thank the reviewer for pointing this out, as we also noticed other methods have been proposed these days. We consider both CoT-Valve and O1-Pruner as concurrent works that have not undergone peer review. Furthermore, both CoT-Valve and O1-Purner are offline methods while Hawkeye is an online methods that could be deployed easily. For example, CoT-Valve requires a complex setup for pairing long-form "standard CoT" chains with shorter CoT variants. This process, combined with LoRA or distillation, enables the model to first learn a shortened CoT representation, making the overall pipeline even more complex. O1-Pruner relies on PPO, where hyperparameters such as the λ value, the number of samples 𝐾, and the PPO clipping range significantly fluctuate the results. Its offline training demands tens of thousands of CoT examples. In contrast, Hawkeye performs GRPO-based online learning and can be finished with only 2k samples. Hawkeye can achieve better token saving and compression rate than O1-Pruner by using a simpler reward function, fewer samples, and smaller models.
> For the sake of completeness, we provide a comparison based on the results they reported on the drafts. Since the models and configurations are different, we care about compression rates and accuracies on the benchmarks. Hawkeye can achive better compresstion ratio compared with O1-Pruner and similar value on CoT-Valve with a 7B model. Commonly, larger models have is more potential to have better space to get compressed. We have confidence that Hawkeye can perform bettwe than the baselines with the same model.
>
> **CoT-Valve VS Hawkeye**:
> | Method | CoT-Valve | Hawkeye |
> |--------|-----------|---------|
> | Dataset | GSM8k | GSM8k |
> | Model | QwQ-32B-Preview | DeepSeek-R1-Distill-Qwen-7B |
> | Acc | 95.1% to 94% | 90.7% to 88.9% |
> | Tokens | 741.1 to 352.8 | 331 to 171 |
> | Compression Ratio (Higher is better) | 52.4% | 48.3% |
>
> **O1-Pruner VS Hawkeye**
> | Method | O1-Pruner | Hawkeye | Hawkeye |
> |--------|-----------|---------|---------|
> | Dataset | MATH | MATH | MATH500 |
> | Model | QwQ-32B-Preview | DeepSeek-R1-Distill-Qwen-7B | DeepSeek-R1-Distill-Qwen-7B |
> | Acc | 90.6% to 91.0% | 92.3% to 89.4% | 80.1% to 75.5% |
> | Tokens | 2191 to 1385 | 942 to 361 | 1463 to 474.9 |
> | Compression Ratio (Higher is better) | 36.7% | 61.7% | 67.5% |
>
> **4. The distinction between HAWKEYE (Full) and HAWKEYE (CoT-only)**: The tokens generated by Hawkeye can be divided into the thinking tokens enclosed between [Think] and [/Think], and the the response tokens. Hawkeye (full) refers to the complete reasoning process. It includes both the Hawkeye CoT reasoning (i.e., the [Think]...[/Think] content generated by the large model) and the responses generated by the small model.
>
> We will incorporate this clarification into the paper to help readers better understand the structure of the generated outputs.
>
> Once again, we sincerely thank you for your valuable review. We hope that the above responses sufficiently address your concerns :)

---

> ### Comment · Reviewer_rwTc · 2025-06-07
> **Response**
>
> I sincerely thank the authors for their detailed responses. Most of my concerns have been solved. I will keep my score.

---

> > ### Author Response · Authors · 2025-06-11
> >
> > Thank you very much for taking the time to read our responses and for acknowledging that your concerns have been addressed. We truly appreciate your feedback and engagement throughout the review process.

---

### Decision · Program_Chairs · 2025-07-08

**Decision:**

Accept

**Comment:**

The paper introduces a novel model-collaboration scheme to first train the large models to generate concise instructions, then a small model completes the response in a more human-readable way. The proposed approach can achieve very high token efficiency while almost maintaining the accuracy over a wide set of reasoning benchmarks.

Pros:
- The overall proposed framework is novel, and the results are comprehensive over a wide set of benchmarks.
- The set up is reasonable and is an interesting way to utilize large and small models' advantages respectively.

Cons:
- Some reviewers point out issues with clarity of the paper (e.g., clarify thinking / response). The authors should take these points into consideration for the final version to make the paper easier to understand.
- Some reviewers point out a lack of comparisons to other token efficiency baselines like CoT-Valve and O1-Pruner. The authors added results in the rebuttal and are encouraged to include those results in the final version.

Overall I would recommend acceptance based on the novel insights and comprehensive results.